# Decreased Visual Search Behavior in Elderly Drivers during the Early Phase of Reverse Parking, But an Increase during the Late Phase

**DOI:** 10.3390/s23239555

**Published:** 2023-12-01

**Authors:** Siyeong Kim, Ken Kondo, Naoto Noguchi, Ryoto Akiyama, Yoko Ibe, Yeongae Yang, Bumsuk Lee

**Affiliations:** 1Doctor’s Program, Graduate School of Health Sciences, Gunma University, Maebashi 371-0044, Japan; kimsiyeongkor@gmail.com; 2Department of Occupational Therapy, Faculty of Rehabilitation, Gunma Paz University, Takasaki 370-0006, Japan; kenkondoot@gmail.com; 3Graduate School of Health Sciences, Gunma University, Maebashi 371-0044, Japan; noguchinaoto@gunma-u.ac.jp (N.N.); akiyamaryoto@gunma-u.ac.jp (R.A.); 4Department of Rehabilitation, Gunma University Hospital, Maebashi 371-8511, Japan; yokofuka@gunma-u.ac.jp; 5College of Biomedical Science and Engineering, Inje University, Gimhae-si 50834, Republic of Korea; otyya62@inje.ac.kr; 6Institute of Aged Life Redesign, Inje University, Gimhae-si 50834, Republic of Korea

**Keywords:** elderly driver, visual search behavior, eye tracker, reverse parking

## Abstract

The aim of this study was to assess the characteristics of visual search behavior in elderly drivers in reverse parking. Fourteen healthy elderly and fourteen expert drivers performed a perpendicular parking task. The parking process was divided into three consecutive phases (Forward, Reverse, and Straighten the wheel) and the visual search behavior was monitored using an eye tracker (Tobii Pro Glasses 2). In addition, driving-related tests and quality of life were evaluated in elderly drivers. As a result, elderly drivers had a shorter time of gaze at the vertex of the parking space both in direct vision and reflected in the driver-side mirror during the Forward and the Reverse phases. In contrast, they had increased gaze time in the passenger-side mirror in the Straighten the wheel phase. Multiple regression analysis revealed that quality of life could be predicted by the total gaze time in the Straighten the wheel phase (*β* = −0.45), driving attitude (*β* = 0.62), and driving performance (*β* = 0.58); the adjusted *R*^2^ value was 0.87. These observations could improve our understanding of the characteristics of visual search behavior in parking performance and how this behavior is related to quality of life in elderly drivers.

## 1. Introduction

Reverse perpendicular parking is a type of parking in which the vehicle enters the parking space from the rear at a 90-degree angle. This parking type has the advantage of higher spatial efficiency compared to parallel or angle parking [1]. On the other hand, it is known that parking in reverse increases the risk of accidents. The American Automobile Association reported that approximately 25% of all vehicle crashes occurred during reverse maneuvers [2]. Previous studies have suggested that the increased risk of accidents is related to the driver’s visual information processing during backward maneuvering. At first, the amount of visual information on the heading direction critically decreases because the information can be only acquired through mirrors and shoulder checks [3]. In addition, drivers need to pay attention in all directions to avoid collisions with other vehicles or pedestrians [4]. In backward maneuvering, moreover, the movement of the front of the vehicle is opposite to the direction of the steering wheel; consequently, it often causes left–right confusion.

Visual impairment is related to poor driving performance. For example, the shrinkage of the field of view decreases the amount of information available for visual searching, leading to difficulties in vehicle positioning, a longer duration of maneuvering, and more frequent trajectory adjustments [5]. Impaired dynamic visual acuity results in a high incidence of accidents and convictions [6]. Moreover, lower contrast sensitivity is related to impaired traffic sign-reading accuracy on roads [7].

In addition to the visual impairment itself, visuospatial information processing is also a critical factor in safe driving [8]. Generally, when drivers sit in vehicles, they need to extend their peripersonal space to the vehicle’s size in order to integrate the vehicle into their body scheme [9]. Visual feedback helps with peripersonal space representation in a dynamic environment [10]. In addition, recent studies utilizing an eye tracker have revealed how these visuospatial information processes are represented in gaze behavior [11]. For instance, drivers regularly look far ahead to make a visuospatial reference point, which is called a future path, for planning a trajectory in the traveling direction [12,13,14]. When approaching a curve, they make a reference point called a tangent point on the inside of the curve 1–2 s before turning the steering wheel [15,16,17,18]. One study indicated that 94% of drivers gazed at the tangent point on a curved road [19]. In contrast, reduced visual searching for the tangent point resulted in unstable line-keeping behavior [20], and increased time taken to complete the course [21,22].

The research gaps of this study include the following:Although the importance of visuospatial information processing in road driving is widely recognized, there is limited evidence during reverse parking. It is still unknown whether visuospatial information processing is presented during parking and how it contributes to backward maneuvering.It has already been reported that elderly drivers have less visual search behavior [23,24], and this decreased behavior is related to the risk of accidents during backward maneuvering [25]. The risk directly or indirectly leads to driving cessation and decreased out-of-home activity levels [26]. However, the contribution of visual search behavior in reverse parking to parking performance as well as quality of life (QOL) in elderly drivers has not been adequately discussed.

The main contributions of this study include the following:To identify the visual search behavior characteristics of elderly drivers during reverse parking and compare these characteristics between elderly and expert drivers.To clarify how the characteristics of visual search behavior in reverse parking contribute to parking performance as well as QOL in elderly drivers.

The findings of this study may provide the fundamental basis for building specific strategies for improving parking performance, analyzing visual search behavior during automated driving, and understanding the role of visual search behavior in the maintenance of driving and QOL in elderly drivers.

The rest of this paper is organized as follows: Section 2 describes our research approach (Figure 1), including a parking task and the eye-tracking system. Section 3 presents the results of visual search behavior during the parking task and its relationship with parking performance and QOL in elderly drivers. Finally, the discussion is presented in Section 4.

## 2. Materials and Methods

### 2.1. Participants

Fourteen healthy elderly drivers (2 males and 12 females, 73.7 ± 3.4 years) who visited the local driving school to renew their licenses were recruited. The inclusion criteria were (1) holding a valid driver’s license, (2) being without eye diseases, and (3) having the ability to follow commands. The exclusion criteria were (1) a clinical diagnosis of dementia and (2) having physical issues that affect driving. To identify the characteristics of visual search behavior in elderly drivers, data on 14 expert drivers (12 males and 2 females, 42.1 ± 11.4 years) were also collected. This study was approved by the University Ethical Review Board for Medical Research Involving Human Subjects.

### 2.2. Reverse Perpendicular Parking Task

The parking task was conducted at a parking space in a right-hand-drive vehicle equipped with double pedal devices on the passenger side. The elderly participants sat in the driver’s seat and were asked to adjust the seat position and mirrors, and instructed to drive to the parking area, enter the parking space, and then, return to the starting point. A driving instructor sat in the passenger seat to evaluate parking performance. Expert drivers conducted the same task without the evaluation of parking performance.

As shown in Figure 2, the reverse parking process was divided into three consecutive phases in accordance with a previous study [3]: Forward (approach and pass the parking space), Reverse (shift the gear into reverse, turn the steering wheel to the right and start to move backward) and Straighten the wheel (turn the steering wheel to the left and complete the parking).

### 2.3. Eye-Tracking System

An eye-tracking device (Tobii Pro Glasses 2, Tobii Technology Inc., Danderyd, Sweden) was used to measure participants’ visual search behavior during the parking task. The device was designed with a high-precision, ultra-light, and non-invasive head-tracking module that ensures comfort and freedom of behavior [27]. The driver’s visual search behaviors were monitored by a camera with a resolution of 1920 × 1080 at 25 fps and a sampling rate of 50 Hz. The camera has a 90-degree field of view and can record videos during the task with a range of 52 degrees and 82 degrees in the horizontal plane and the vertical plane, respectively. The eye-tracking device is widely used to assess gaze behaviors in real-world driving [28,29,30]. Recent eye-tracking studies have identified the role of look-ahead fixation in road safety [31,32,33] and how the device can be employed to assess cognitive load [27,34] and driving fatigue [35]. After completing the task, we used Tobii Pro Lab Software (version 1.171) to analyze the drivers’ eye movement during the task.

Based on the concepts of previous studies [15,21] and our pilot data, four parameters indicating visuospatial information processing during reverse parking were identified (Figure 3): horizontal scanning (a, a sequence of horizontal gaze shifting on the vehicle bonnet), vertex of the parking space (b, a group of gaze fixations on the vertex of the parking space), reciprocating scanning (c, a sequence of gaze shifting between the participant’s own vehicle and objects in the mirror), and sawtooth pattern (d, a sequence of gaze shifting between inside and outside the mirror).

### 2.4. Conventional Tests

Elderly drivers’ driving performance was evaluated using six items by a single driving instructor: S-shape curve, changing direction, left turn, right turn, curve driving, and control of the steering wheel. The score ranges from 1 to 4 points: 1 = driving was unsafe, 2 = exhibited a couple of hazardous maneuvers, 3 = exhibited a few minor driving errors, and 4 = no obvious driving errors. The minimum total score is 6 and the maximum is 24.

Driving attitude was assessed using two items: driving attitude (i.e., I like driving) and driving confidence (i.e., I am confident in my driving). The score ranges from 1 to 4 points: 1 = strongly disagree, 2 = disagree, 3 = agree and 4 = strongly agree. The minimum total score is 2 and the maximum is 8.

QOL was assessed using WHOQOL-26. WHOQOL-26 assesses QOL in four domains including physical health, psychological, social relationships, and environment. Although the instrument captures multiple aspects of QOL, we only used the environmental domain as our interest was focused on the relationships between driving and QOL [36,37]. The environmental health domain covers issues related to financial resources, safety, health and social services, living physical environment, opportunities to acquire new skills and knowledge, recreation, leisure, and transportation [38].

### 2.5. Statistical Analysis

The assumption of distributional normality was tested using the Shapiro–Wilk test, and we found that the data were not normally distributed. As a result, non-parametric statistics were applied. Differences between the elderly and expert drivers were examined using the Mann–Whitney test. Associations between visuospatial parameters and the number of attempts to park by the elderly driver were examined via Spearman’s rank correlation.

As our research aim was to know the contribution of visual search behavior in reverse parking, we selected visuospatial parameters, driving performance, and driving attitude as independent variables, and the environmental domain of WHOQOL-26 as the dependent variable. The analysis, therefore, was performed via a forced-entry linear multiple regression analysis [39,40]. The statistical software SPSS ver.29.0 J for Windows (SPSS Japan, Tokyo, Japan) was used for the analysis. Values of *p* < 0.05 were considered significant.

## 3. Results

Table 1 shows the results of the comparison between expert and elderly drivers. The median age and ratio of females were higher in elderly drivers.

Figure 4 shows the visual search behaviors of the representatives of each group. The red line indicates the sequence of gaze points, and the red circle indicates the last gaze point. The expert driver gazed at the vertex of the parking space through the windshield when approaching the parking space (Figure 4(Aa)), and gazed again at the vertex through the driver-side mirror after passing the parking space (Figure 4(Ab)). Reciprocated scanning between the vehicle and the vertex was observed in the mirror, implying object identification and distance estimation to avoid a collision (Figure 4(Ac)). When the parking task was almost completed, the sequence of gaze points on the bonnet was observed, implying visual search behavior for returning to the driving course (Figure 4(Ad)). On the other hand, in the elderly driver, horizontal scanning on the bonnet was observed when approaching the parking space (Figure 4(Ba)). After the vehicle passed the parking space, the gaze points relatively stayed outside the mirror (Figure 4(Bb)), and similar searching behavior was observed in the Reverse phase as well (Figure 4(Bc)). However, when the parking task was almost completed, a group of gaze points was found in the passenger-side mirror, implying a compensatory strategy to counterbalance the insufficient visual scanning in the previous phases (Figure 4(Bd)).

Table 2 shows comparisons between the elderly and expert drivers. The elderly drivers had a shorter time of gaze at the vertex of the parking space both in direct vision and reflected in the driver-side mirror during the Forward and Reverse phases. Similarly, the total gaze time of horizontal scanning and reciprocating scanning in the Reverse phase was shorter than those of expert drivers. In contrast, they had increased gaze time in the passenger-side mirror in the Straighten the wheel phase. The time to complete parking and the number of attempts to park were higher in elderly drivers.

The results of correlations between the total gaze time and the number of attempts to park in the elderly drivers are shown in Table 3. Only the total gaze time in the Forward phase was negatively correlated with the number of attempts to park.

The multiple regression model identified that QOL could be predicted by the total gaze time in the Straighten the wheel phase (*β* = −0.45, *p* < 0.02), driving attitude (*β* = 0.62, *p* < 0.01), and driving performance (*β* = 0.58, *p* < 0.01); the adjusted *R*^2^ value was 0.87 (Table 4). It is worth noting that the total gaze time in the Straighten the wheel phase negatively affected QOL.

## 4. Discussion

We found that the visual search behavior while parking was different between the two groups. Elderly drivers had a shorter time of gaze at the vertex of the parking space both in direct vision and reflected in the driver-side mirror during the early phases. In contrast, they had increased gaze time in the passenger-side mirror in the late phase. Moreover, multiple regression analysis revealed that QOL could be predicted by the lower total gaze time in the late phase, implying that visual search behavior is related to QOL in elderly drivers.

Significant differences between groups in visuospatial parameters differed in each phase of the parking process. This observation is consistent with a previous study that compared visual scanning in the intersection. The study found that elderly drivers had a lower proportion of time scanning to the right than middle-aged drivers while approaching and exiting the intersection, although there was no age difference while approaching the median [41]. Another study reported a similar situation-dependent discrepancy. Kunishige et al. compared eye movements in simulated driving between elderly and young adults and found that the number of saccades was different only in turn, not in lane change [42]. These discrepancies could reflect the dynamic perceptual–cognitive mechanisms in driving. Indeed, our parking task included moving forward/backward and multiple direction changes of the steering wheels. Based on previous studies and our findings, it would be reasonable to suggest that phase-specific visual search behavior needs to be considered in the evaluation of driving performance in elderly drivers.

Elderly drivers gazed significantly less at the vertex of the parking space than expert drivers. The vertex of the parking space may play the role of the tangent point in our experiment. Previous studies suggested that the tangent point provides the driver with a visual pivot for planning a trajectory in the traveling direction, and the inside of the curve plays the role of a tangent point on curved roads [43]. We assume that gazing at the vertex on the driver’s side could be an ideal pivot for calculating the spatial relationship between the vehicle and the parking space in reverse perpendicular parking. In addition, it is worth noting that the identification of the vertex was found not only in direct vision but also reflected in the driver-side mirror. The roles of the mirror are generally considered to be locating, detecting, and identifying objects around a vehicle [44]. We found another critical role of the mirror: to identify the tangent point for guiding forward/backward movement and multiple direction changes of the steering wheel in reverse perpendicular parking (Figure 5).

Considering the negative correlation between the total gaze time in the Forward phase and the number of attempts to park in the elderly drivers, the active visual search behavior in the earlier approach may contribute to better parking performance. According to a previous study, appropriate preliminary behavior in perpendicular parking improves the performance of the final alignment to the parking space [45]. Skilled drivers gaze more at the parking place to prepare their next motion, even when the vehicle is not approaching the place [4]. On the other hand, elderly drivers look less in appropriate directions than young drivers at the start of backing-up [46]. These observations imply that preparatory visual search behavior plays an important role in efficient parking. The advantage of active visual search behaviors in the early phases can be explained by Michon’s driving model [47]. According to the model, a driver’s behavioral strategy is divided into three hierarchical levels: strategy, maneuvering, and control. Searching for and analyzing visual clues from the environment at the strategy level is important for planning the maneuvering and control levels [48] and influences overall performance [45]. When applying the model to the present study, poor performance in the final phase and subsequent additional attempts to park can be explained by reduced visual search behavior while approaching the parking space. Although this finding does not allow us to discuss the effectiveness of interventions, we assume that instructing drivers to facilitate their visual attention in the early phases may be useful for improving their parking performance (Figure 6).

Another meaningful difference was found in the parameter of the sawtooth pattern. The sawtooth pattern indicates a reciprocation pattern of oculomotor movement [49,50]. The pattern is generated when vision guides the movement of the steering wheel. For example, drivers look first at a point in the world 1–3 s in the future; then, they track this point for around 0.4 s before generating a new waypoint on the future path, and repeat the pattern regularly [51]. Contrary to our expectation, however, a longer time of the sawtooth pattern was found in elderly drivers. This prolonged time of gaze in the elderly drivers is consistent with a previous study that reported that elderly drivers look more at lines and markings on the road than younger drivers to position themselves [52]. These additional gaze behaviors are considered a compensatory mechanism for supporting safe driving in elderly drivers [24]. The compensatory mechanism is often considered a protective mechanism in aging and car driving. For example, in the model of the successful aging model suggested by Baltes and Graf, it is assumed that elderly drivers bring in additional resources to compensate for age-related degradation [53]. This assumption is supported by our finding that a difference between the two groups was only found in the passenger-side mirror in the Straighten the wheel phase. It is not hard to imagine that the passenger-side mirror does not play an important role during the Forward and Reverse phases in our experiment design. Elderly drivers may perform compensatory visual search behavior to obtain additional information in the late phase of parking to guarantee safe parking (Figure 7).

The multiple regression model identified that QOL can be predicted using the total gaze time in the Straighten the wheel phase, driving attitude, and driving performance. The positive influence of driving attitude and performance on QOL and the negative influence of driving cessation on general health conditions and QOL are widely acknowledged [48,54,55,56]. On the other hand, studies on the influence of visual function on QOL are limited. It is only reported that vision difficulties and distant/near-vision impairments have significant impacts on QOL [36,37]. Interesting findings of the present study are that visual search behavior is also related to QOL, and that behavior in the late phase could negatively affect QOL. As suggested above, active visual search behavior in the late phase of parking is considered a compensatory mechanism for supporting safe driving. On the other hand, it is not recommended to generalize our finding to the universal relationship between visual function and QOL. Although the greater part of the information used in driving is visual, the precise percentage attributable to vision is under discussion [57,58], not to mention the contribution of driving to QOL [59,60]. Nevertheless, in spite of these uncertainties, considering that QOL in this study was an environmental domain that included the capability of the transportation, and that the participants were elderly people who wished to continue driving, it would be a reasonable assumption to consider that visual search behavior could affect QOL to some extent.

Several limitations should be acknowledged. Firstly, the parking space was always located on the driver’s side in this study, and therefore, the influence of the approach side was not considered. When turning at an intersection, the approach side between the driver and passenger sides could make differences in driving performances, such as speed variation or the stability of vehicle position [61]. Visual search behavior could be also influenced by the approach side. Similarly, the driver’s behavior can be influenced by the side of the road traffic drives on. Our study was only conducted in a right-driving country, and it is unknown whether the same results would be obtained from a left-driving country. Future studies in left-driving countries may clarify the influence of driving side on visual search behavior. Secondly, the experiment was conducted in a risk-free setting. In an actual parking environment, pedestrians or other vehicles may exist around the parking space; therefore, investigating gaze under realistic parking conditions needs to be considered. Thirdly, the gender balance between the two groups was very unequal in this study. Several studies have indicated that driving behaviors were influenced by gender characteristics [62,63]. Although we did not consider gender differences due to the small sample in this study, future studies including a large sample may clarify the relationship between gender characteristics and the visual search behavior of elderly drivers in reverse parking. Lastly, the participants drove our standard vehicles, not their own vehicles. In general, elderly people have a limited capacity to adapt to new environments. We were not able to capture the adaptive capacity of the participants. Further studies integrating the above-mentioned limitations may provide new insight for building specific strategies for improving parking performance, analyzing visual search behavior during automated driving, and understanding the role of visual search behavior in the maintenance of driving and QOL in elderly drivers.

## 5. Conclusions

There are two main contributions of this research:Elderly drivers had a shorter time of gaze at the vertex of the parking space both in direct vision and reflected in the driver-side mirror during the early phases, resulting in poor parking performance. On the other hand, they increased their gaze time in the passenger-side mirror in the late phase, implying that a compensatory strategy was used to counterbalance the insufficient visual scanning in the previous phases.The lower total gaze time in the late phase is related to QOL. This observation could improve our understanding of the characteristics of visual search behavior in parking performance and how this behavior is related to QOL in elderly drivers.

## Figures and Tables

**Figure 1 sensors-23-09555-f001:**
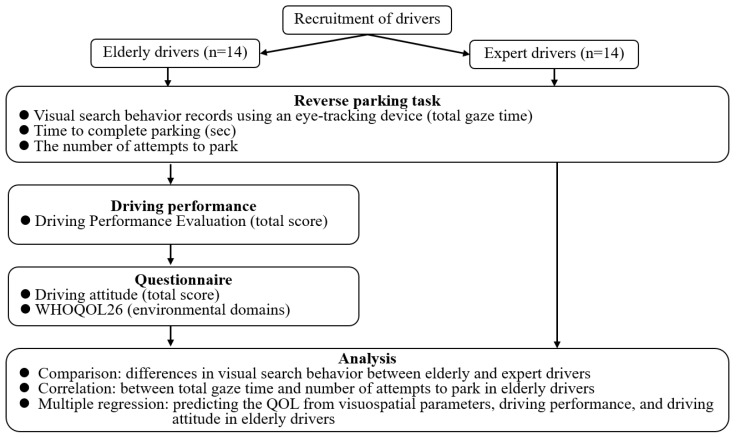
A flowchart of the experiment.

**Figure 2 sensors-23-09555-f002:**
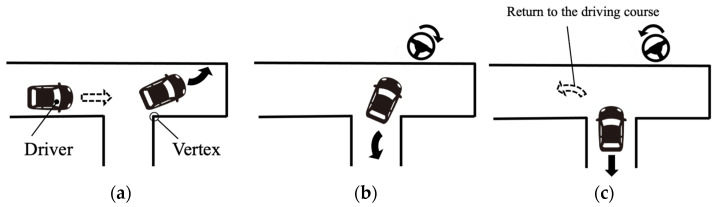
Three phases of the perpendicular parking in the experiment: Forward (**a**), Reverse, (**b**) and Straighten the wheel (**c**). The driver approaches and passes the parking space (**a**), shifts the gear into reverse and starts to move backward (**b**), and straightens the wheel (**c**).

**Figure 3 sensors-23-09555-f003:**
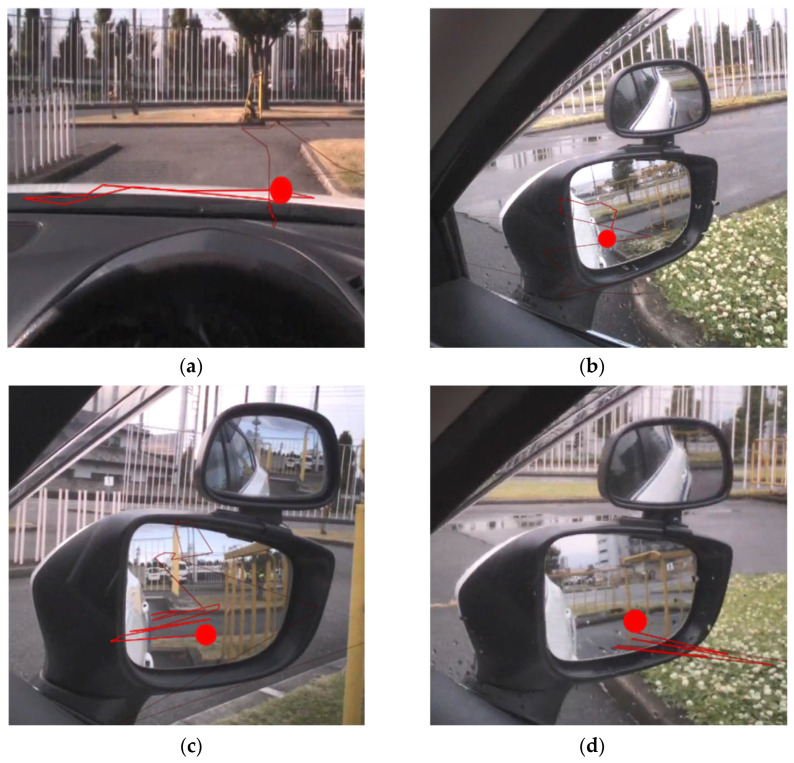
Visuospatial parameters indicating visuospatial information processing. The red line indicates the sequence of gaze points, and the red circle indicates the last gaze point. Horizontal scanning (**a**): horizontal gaze tracking on the bonnet. Vertex of the parking space (**b**): a group of gaze points on the vertex of the parking space. Reciprocating scanning (**c**): reciprocating movements of gaze points between the vehicle and an obstacle. Sawtooth pattern (**d**): reciprocating movements of gaze points between inside and outside the mirror.

**Figure 4 sensors-23-09555-f004:**
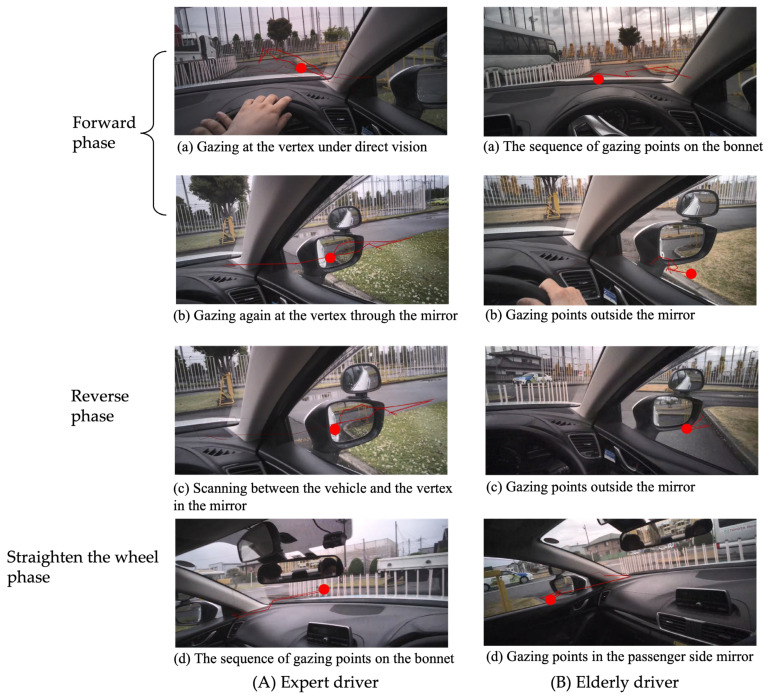
The visual search behavior in reverse parking of the representatives of each group. The red line indicates the sequence of gaze points, and the red circle indicates the last gaze point. (**A**) The expert driver gazed at the vertex of the parking space through the windshield while approaching the parking space (**Aa**) and gazed again at the vertex through the driver-side mirror after passing the parking space (**Ab**). Reciprocated scanning between the vehicle and the vertex of the parking space was observed in the mirror (**Ac**). When the parking task was almost completed, a sequence of gaze points on the bonnet was observed, implying that visual search behavior for returning to the driving course had already started (**Ad**). (**B**) In the elderly driver, a sequence of gaze points was observed on the bonnet when approaching the parking space (**Ba**). After the vehicle passed the parking space, the gaze points relatively stayed outside the mirror (**Bb**), and similar searching behavior was observed in the Reverse phase as well (**Bc**). When the parking task was almost completed, a group of gaze points was found in the passenger-side mirror (**Bd**).

**Figure 5 sensors-23-09555-f005:**
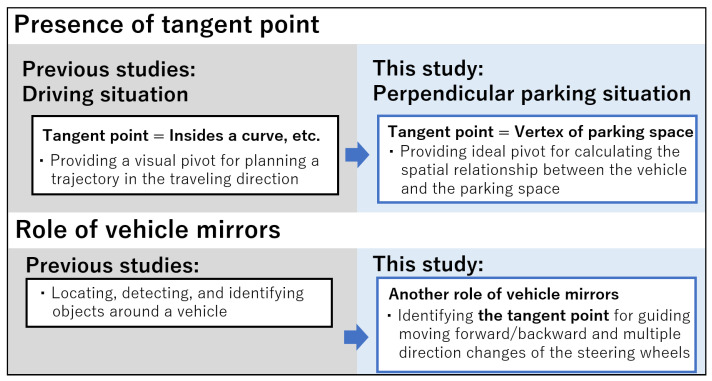
The critical role of the vertex of the parking space as a tangent point in perpendicular parking.

**Figure 6 sensors-23-09555-f006:**
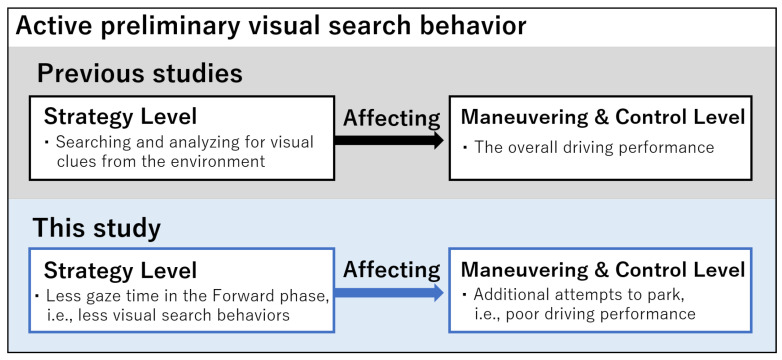
Application of Michon’s driving model: fewer visual search behaviors at the strategy level led to additional attempts to park in the maneuvering and control levels in elderly drivers.

**Figure 7 sensors-23-09555-f007:**
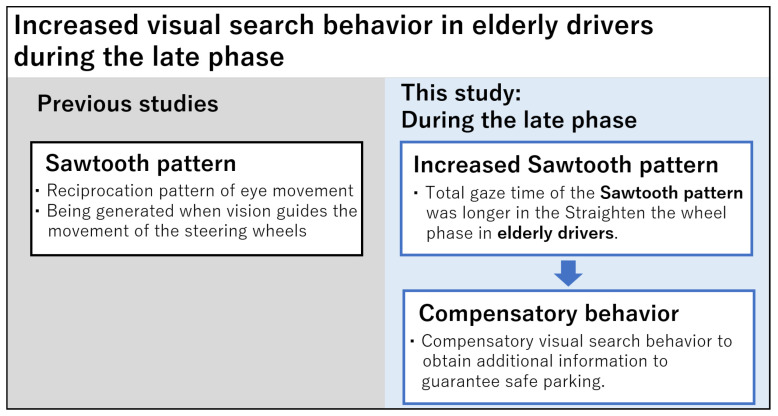
Compensatory visual search behaviors: longer gaze time of the sawtooth pattern in elderly drivers.

**Table 1 sensors-23-09555-t001:** Characteristics of two groups.

	Expert Drivers (*n =* 14)	Elderly Drivers (*n* = 14)	*p*
Age	42.1 ± 11.4	73.7 ± 3.4	**<0.001**
Gender (male/female)	12/2	2/12	**<0.001**
BMI ^1^	23.1 ± 3.2	24.2 ± 3.8	0.53
Years of education		6.0 ± 2.0	
Years of driving license		50.6 ± 6.7	
Years of driving instructor	16.0 ± 9.4		
Driving attitude		6.0 ± 0.9	
Driving performance		18.0 ± 3.1	
WHOQOL environment ^2^		3.7 ± 0.6	

Values in bold indicate significant difference; ^1^ BMI: body mass index, ^2^ QOL: quality of life.

**Table 2 sensors-23-09555-t002:** Comparison of total gaze time and driving performance.

			Expert Drivers	Elderly Drivers	*p*
Total			11,486 ± 5114	9052 ± 7254	0.28
Forward phase	Direct vision	Horizontal scanning	2820 ± 1766	2048 ± 1868	0.18
		Vertex of the parking space	568 ± 782	122 ± 228	**0.02**
	Driver-side mirror	Vertex of the parking space	198 ± 312	46 ± 172	**<0.01**
		Reciprocating scanning	152 ± 316	46 ± 126	0.49
		Sawtooth pattern	508 ± 1030	208 ± 332	0.79
	Passenger-side mirror	Reciprocating scanning	82 ± 304	0 ± 0	0.32
		Sawtooth pattern	74 ± 156	0 ± 0	0.07
Reverse phase	Direct vision	Horizontal scanning	830 ± 864	254 ± 452	**0.02**
	Driver-side mirror	Vertex of the parking space	512 ± 798	38 ± 108	**0.01**
		Reciprocating scanning	1186 ± 898	288 ± 586	**<0.001**
		Sawtooth pattern	1420 ± 1736	1726 ± 2024	0.75
	Passenger-side mirror	Reciprocating scanning	8 ± 32	0 ± 0	0.32
		Sawtooth pattern	772 ± 1066	2022 ± 2904	0.62
Straighten the wheel phase	Direct vision	Horizontal scanning	1280 ± 1098	672 ± 810	0.12
	Driver-side mirror	Vertex of the parking space	28 ± 106	0 ± 0	0.32
		Reciprocating scanning	374 ± 622	198 ± 542	0.25
		Sawtooth pattern	442 ± 634	588 ± 1026	0.94
	Passenger-side mirror	Reciprocating scanning	48 ± 176	56 ± 182	0.55
		Sawtooth pattern	184 ± 558	742 ± 1136	**0.04**
Time to complete parking (s)			41.0 ± 14.8	72.9 ± 52.9	**<0.01**
Number of attempts to park			1.2 ± 0.6	2.6 ± 2.1	**0.01**
					(millisecond)

Values in bold indicate significant difference.

**Table 3 sensors-23-09555-t003:** Spearman’s rank correlation coefficients between the total gaze time in each phase and number of attempts to park.

	Number of Attempts
Total	−0.09
Forward phase	**−0.56**
Reverse phase	−0.09
Straighten the wheel phase	0.29

Values in bold indicate significant correlation (*p* < 0.05).

**Table 4 sensors-23-09555-t004:** Results of forced-entry linear multiple regression analyses of QOL-related variables.

Dependent Variable	Adjusted *R*^2^	Independent Variable	Standardized *β*	*p* Value
QOL Environment	0.87	Total gaze time in Straighten the wheel phase	−0.45	0.02
		Driving attitude	0.62	<0.01
		Driving performance	0.58	<0.01

## Data Availability

The data presented in this study are available on request from the corresponding author. The data are not publicly available due to ethical reasons.

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
