# Peer review of "Decreased Visual Search Behavior in Elderly Drivers during the Early Phase of Reverse Parking, But an Increase during the Late Phase"

_sensors, 2023, doi:10.3390/s23239555_

Round 1

Reviewer 1 Report

Comments and Suggestions for Authors

The study examined the visual search behavior of fourteen elderly and expert drivers across three specific stages of reverse parking, utilizing an eye tracker. These stages comprised forward, reverse, and straightening the wheel maneuvers. Additionally, the authors conducted multiple regression analysis to probe into correlations between participants' quality of life and variables such as total gaze time during the straighten the wheel phase.

While the authors' commendable use of advanced technology led to intriguing results in certain aspects, there are four pivotal concerns that require careful attention.

Firstly, the practical implications of the research findings, particularly concerning the disparities in visual search patterns between elderly participants and others, lack clarity. A comprehensive explanation from the authors is essential, especially due to the article's title.

Secondly, the rationale behind considering gaze time in the straighten the wheel phase as an independent variable for predicting quality of life via multiple regression appears ambiguous. It is imperative for the authors to clarify whether any established correlations were identified and modeled.

Thirdly, the measurement of certain variables, such as driving attitude, remains unclear. While various methods exist to measure driving attitude, your explanation and chosen method lack clarity and detail.

Lastly, a more in-depth exploration of the existing literature could significantly enhance the overall paper. Therefore, I recommend that the authors conduct a more thorough review of pertinent literature to bolster the scholarly foundation of their work.

Author Response

Thank you very much for your comments and suggestions. In separate transmittal letters, we summarized specific responses to all the queries. 

Reviewer 2 Report

Comments and Suggestions for Authors

Dear Authors,

The article is very interesting and I enjoyed reading its content. The proportion of drivers compared to 12 older women and 12 young, experienced men is a bit surprising. But I have no objections to this. I congratulate the Authors on the research idea.

However, there’s some details in this paper may deserve further review and carefully check. Here are some examples for reference.

General Comments and Suggestions for Authors

1.       Lines 60–31 – please edit it so that it leaves a stylistic research gap.

2.       Lines 68 – 70 please edit stylistically so that it constitutes the purpose of the article.

3.       At the end of the Introduction, please provide a flow chart with a step-by-step description of the research method used in this article. And indicate where in the flow chart the analysis related to the basic research question begins.

4.       At the end of the Introduction, in accordance with the guidelines for authors, please give: Finally, briefly mention the main aim of the work and highlight the principal conclusions.

5.       In lines 220-231. Please provide diagrams illustrating the text for this fragment. It would be very interesting and informative.

6.       Similarly, please add explanatory diagrams illustrating the text to lines 232–250.

7.       Similarly, please add explanatory diagrams illustrating the text to lines 2512–266.

8.       The observation described in lines 277–279 is probably much exaggerated regarding low QOL. This was not proven in this study.

9.       Conclusion in lines 290–293 can also be used to design automatic guidance devices in cars during parking.

10.    Please expand and complete your applications. Who will benefit most from the results of this study? Who exactly are they intended for? Will research in other countries obtain the same results in similar studies?

Detailed Comments and Suggestions for Authors

11.    According to the instructions for authors, please place Figure 1 on line 97.

12.    Please match the diagrams and names correctly in Figure 1A. There are 4 pictures in part A and 3 names: a, b and c. Please correlate them.

13.    Please enlarge the photos in Figure 1B so that they cover the entire width of the A4 text. What do the authors mean by the term vertex? Is this the middle of a parking space? Is it a curb? Please describe it in the text or draw it in Figure 1.

14.    It would be good to extend the photographs in Figure 2B to the entire width of the A4 text.

15.    Under Table 2. What does offset (ms) mean?

16.    In the name Table 3. What does (rs) mean?

Author Response

(The authors gave the same response as above.)

Reviewer 3 Report

Comments and Suggestions for Authors

This paper presents an interesting study. It is overall appropriately written and presented.

A few remarks:

- There are some issues with the experimental groups. First, the number of participants in each group is small (only 14); secondly, the gender balance is very different in the two groups, and this may introduce confounding effects, with observed differences potentially being partly related to gender rather than age. The author should discuss these possible limitations to the validity of their study.

- Was the driving instructor always the same person? Why was no evaluation given on expert drivers? It would have been a useful additional information for comparing the two groups.

- I am familiar with the device used, but it was not presented very well to the readers. More information on the technical characteristics should be given. A short review on the use of eye-trackers in road safety should also been included, referring to more recent works (many of the cited references are quite old); this would underline that the instrument used is extensively adopted in this line of research. A few examples: https://doi.org/10.1145/3588015.3589512, https://doi.org/10.1016/j.trpro.2023.02.180, https://doi.org/10.1016/j.trf.2023.10.014, https://doi.org/10.1016/j.trf.2023.09.016, https://doi.org/10.1016/j.trf.2022.11.013, https://doi.org/10.1016/j.trf.2023.01.021.

- I found very interesting the investigation of the relationship with QOL. However, more detail should be given on the WHOQOL 26; maybe a specific sub-section should be dedicated to that.

- The discussion is overall well-written, except for the part regarding the relationship with QOL, which is not completely clear. I believe this part could be extended.

- I suggest using some bullet points to better summarize the conclusions.

- I personally do not like having figures with two sub-levels. I suggest separating Figure 1A from 1B, and 2A from 2B

Comments on the Quality of English Language

No major issues

Author Response

(The authors gave the same response as above.)

Round 2

Reviewer 1 Report

Comments and Suggestions for Authors

No comment

Author Response

Thank you very much for your review. 
In separate transmittal letters, we summarized specific responses to all the queries. We hope that the manuscript has been improved and will now be considered suitable for publication in Sensors.

Reviewer 3 Report

Comments and Suggestions for Authors

The authors have invested significant effort in enhancing the manuscript. I have a few additional minor suggestions to further refine the final version:

* The limitations of the study are typically included in the Conclusion section rather than the Discussion. 

* Providing clearer references to the future development of this research would enhance the manuscript. 

* While the authors have partially addressed my previous comment on the eye-tracker by incorporating technical details, a brief overview of recent works utilizing eye-tracking devices in road safety is still missing. It would be valuable to demonstrate to readers, especially those unfamiliar with the tool, that it can be employed to assess cognitive load, driving fatigue, and look-ahead fixations. The authors may refer to my earlier review for paper suggestions, though they are encouraged to explore the literature further. 

* In reference 57, the names and surnames are inverted. It should read as follows: Cordellieri, P., Baralla, F., Ferlazzo, F., Sgalla, R., Piccardi, L., & Giannini, A. M. (2016). Gender effects in young road users on road safety attitudes, behaviors, and risk perception. Frontiers in psychology, 7, 1412.

 * At line 61, it should be “:” instead of “;”.

Author Response

(The authors gave the same response as above.)
